**www.cambridge.org/ext**

# *This is the way the world ends; not with a bang but a whimper*: Estimating the number and ongoing rate of extinctions of Australian non-marine invertebrates

arthropod; conservation; ghost extinctions; insect; taxonomic bias

**Corresponding author:**
John C.Z. Woinarski;
Email: john.woinarski@cdu.edu.au

John C.Z. Woinarski[1] , Michael F. Braby[2,3], Heloise Gibb[4], Mark S. Harvey[5], Sarah M. Legge[1,6], Jessica R. Marsh[7,8], Melinda L. Moir[9], Tim R. New[10], Michael G. Rix[11] and Brett P. Murphy[1]

[1]Research Institute for the Environment and Livelihoods, Charles Darwin University, Casuarina, NT, Australia; [2]Division of Ecology and Evolution, Research School of Biology, The Australian National University, Acton, ACT, Australia; [3]Australian National Insect Collection, Canberra, ACT, Australia; [4]Centre for Future Landscapes, La Trobe University, Bundoora, VIC, Australia; [5]Western Australian Museum, Welshpool DC, WA, Australia; [6]Fenner School of Society and the Environment, The Australian National University, Canberra, ACT, Australia; [7]Harry Butler Institute, Murdoch University, Murdoch, WA, Australia; [8]School of Biological Sciences, Faculty of Sciences, Engineering and Technology, The University of Adelaide, Adelaide, SA, Australia; [9]Department of Primary Industries and Regional Development, South Perth, WA, Australia; [10]Department of Environment and Genetics, La Trobe University, Bundoora, VIC, Australia and [11]Queensland Museum, Hendra, QLD, Australia

## Abstract

Biodiversity is in rapid decline, but the extent of loss is not well resolved for poorly known groups. We estimate the number of extinctions for Australian non-marine invertebrates since the European colonisation of the continent. Our analyses use a range of approaches, incorporate stated uncertainties and recognise explicit caveats. We use plausible bounds for the number of species, two approaches for estimating extinction rate, and Monte Carlo simulations to select combinations of projected distributions from these variables. We conclude that 9,111 (plausible bounds of 1,465 to 56,828) Australian species have become extinct over this 236-year period. These estimates dwarf the number of formally recognised extinctions of Australian invertebrates (10 species) and of the single invertebrate species listed as extinct under Australian legislation. We predict that 39–148 species will become extinct in 2024. This is inconsistent with a recent pledge by the Australian government to prevent all extinctions. This high rate of loss is largely a consequence of pervasive taxonomic biases in community concern and conservation investment. Those characteristics also make it challenging to reduce that rate of loss, as there is uncertainty about which invertebrate species are at the most risk. We outline conservation responses to reduce the likelihood of further extinctions.

## Impact statement

A fundamental objective of biodiversity conservation is to prevent extinctions. However, conservation efforts have characteristically been biased towards iconic and well-known taxonomic groups, often at the expense of poorly known taxa, such as most invertebrates. To redress such a narrow perspective, we attempt to estimate the number of extinctions of Australian endemic invertebrates, and to predict the likely number of such extinctions in 2024, explicitly noting caveats in this assessment. Whereas only one invertebrate species is listed as extinct under Australian environmental legislation, we estimate that there have been ~9,000 extinctions (plausible bounds of 1,465 to 56,828) of endemic non-marine invertebrate species since the European colonisation of Australia, a tally that vastly exceeds (by about two orders of magnitude) the number of formally listed extinctions of all Australian biodiversity. Many of these are likely 'ghost extinctions', the loss of undiscovered species that have left no trace. We predict the extinction in 2024 of 39–148 Australian endemic non-marine invertebrate species. With a plausible rate of 1–3 extinctions of Australian invertebrates per week, a recent pledge by the Australian government to prevent any further extinctions is clearly not being met and can only be addressed if highly imperilled invertebrates are recognised and supported. The ongoing loss of so many invertebrate species has probably led to subversion of ecological health and processes, the impacts of which are likely to become increasingly consequential.



## Introduction

Some extinctions are momentous. The loss of the iconic thylacine, *Thylacinus cynocephalus*, Australia's largest marsupial predator and sole recent species in the family Thylacinidae, has been

widely mourned and recognised as a touchstone of biodiversity loss and the need for more effective conservation efforts in Australia (Holmes and Linnard, 2023). In other cases, extinction represents a specific conservation failure: attempts had been made to prevent it, but were unsuccessful for various reasons (Woinarski et al., 2017; Woinarski, 2018). However, many other extinctions occur largely unrecognised, with no targeted efforts made to prevent them, or without knowledge that the species was even in peril or, in some cases, without knowledge that the species even existed ('dark extinctions') (Boehm and Cronk, 2021).

A recent study reported that 97 plant and animal species have been formally listed as extinct in Australia since its European colonisation in 1788, with a further three species listed as extinct in the wild (Woinarski et al., 2019). That tally included 10 invertebrate species, only one of which is listed as extinct under Australian environmental legislation (the Lake Pedder earthworm, *Hypolimnus pedderensis*). However, that study noted that this was likely to be a considerable underestimate of the actual number of invertebrate extinctions. Such underreporting of invertebrate extinctions is a global characteristic and concern (e.g., Dunn, 2005; Carlton, 2023). In Australia, as is the case globally, this underreporting of invertebrate extinctions is largely because of major knowledge gaps about their existence and conservation status (Braby, 2018; Taylor et al., 2018). Such uncertainty is representative, and a consequence, of major biases in conservation concern, with these biases permeating policy and responses (Cardoso et al., 2011a; Walsh et al., 2013) and reflective of societal attitudes that typically favour care for iconic and well-known species, particularly mammals and birds (Tisdell et al., 2006, 2007; Pearson et al., 2022).

In response to escalating rates of biodiversity loss, global initiatives (CBD [Convention on Biological Diversity], 2022) and national policies (Commonwealth of Australia, 2022) have committed to attempt to prevent further extinctions. However, while the Australian government's 2022 commitment aims to prevent any extinction, the global target is much more qualified: "Ensure urgent management actions, to halt human-induced extinction of *known threatened* species …" (emphasis added, Target 4: Kunming-Montreal Global Biodiversity Framework) (CBD, 2022). This latter commitment sidesteps responsibility for trying to prevent the extinction of undescribed species or those not formally listed as threatened. Most invertebrate species will not meet these qualifiers.

There are major challenges in listing, or even estimating the number of, extinctions of invertebrate species (Stork, 2010), and hence of trying to prevent them. A principal obstacle to formally listing invertebrate species as extinct is the evidentiary bar required. The IUCN defines extinction as "there is no reasonable doubt that the last individual has died" and "that exhaustive surveys have been undertaken in all known or likely habitat throughout its historical range" (IUCN Standards and Petitions Subcommittee, 2022). For many invertebrates, such certainty is impossible because of substantial knowledge gaps (Cardoso et al., 2011a). For example, Mora et al. (2011) estimated that only around 14% of all species have been described taxonomically, and Chapman (2009) estimated that only about 30% of Australian invertebrates have been described. The geographic range of many species is unknown, and major problems of detectability for most species, combined with little investment in surveys or robust long-term monitoring, have meant that there are substantial gaps in information on population size and trajectory, rendering quantitative IUCN assessments of their threatened status difficult or impossible under most criteria (Cardoso et al., 2011a; Didham et al., 2020; Rocha-Ortega et al., 2021; Rix et al., 2023). This poor knowledge base creates a data deficiency feedback loop that maintains a cycle of ignorance and inaction (Sanderson et al., 2021). The evidence bar relates not only to the demonstration and formal listing of any species' extinction but also for listing of species as threatened (Moir and Brennan, 2020), such that most highly imperilled Australian invertebrate species are not formally recognised as threatened (New, 2009).

However, many recent studies elsewhere have demonstrated high, and hitherto unrecognised, rates of extinction in at least one large invertebrate group, landsnails (Régnier et al., 2009, 2015a,b), and documented major and ongoing declines across large swathes of the invertebrate fauna (Wagner, 2020; Wagner et al., 2021; Cowie et al., 2022). Such trends are also apparent in Australia, although the evidence is limited (Rix et al., 2017; Braby, 2019; Braby et al., 2021; New, 2022; Monteith, 2023).

One example indicating that there may be many more extinctions of Australian invertebrates than the formally recognised tally is in the fate of the endemic fauna of Christmas Island, an Australian territory in the Indian Ocean; of 200 invertebrate species recognised to be endemic to this island, 49 have not been reported for at least 100 years (James et al., 2019). Although some of these species may have persisted but have not been recorded, undoubtedly many are extinct. For example, in two cases of co-extinction, the flea *Xenopsylla nesiotes* and the tick *Ixodes nitens* were obligate ectoparasites of the extinct Maclear's rat, *Rattus macleari*, endemic to Christmas Island, and have not been recorded since the host's extinction in about 1902 (Colwell et al., 2012; Kwak, 2018). Notwithstanding this evidence, none of the lost Christmas Island invertebrate species are formally recognised as extinct.

The available evidence for assessing extinction rates in invertebrates is thin and inconsistent. In Australia, the number of extinctions is known (with reasonable confidence) for at least one well-studied and comprehensively inventoried group of invertebrates, butterflies. Of 218 Australian endemic species, there have been no known extinctions (Geyle et al., 2021), although the Laced Fritillary, *Argynnis hyperbius inconstans*, an Australian endemic subspecies, is now very likely to be extinct (Lambkin, 2017). Likewise, low extinction rates of butterflies have been reported for other continents (Dunn, 2005). A global review of the conservation status of a large suite of randomly selected Odonata reported that there were zero extinctions amongst the pool of 1,500 species considered (Clausnitzer et al., 2009). However, such apparently low rates of extinction in taxonomic groups with characteristically high dispersal ability contrast to the fates of groups characterised by limited dispersal capability. For example, Sullivan and Ozman-Sullivan (2021) considered that among the world's estimated 1,250,000 mite species, 15% were extinct, with this high rate due to the very small ranges of many species coinciding with high rates of habitat destruction. High rates of extinctions have also been reported for the world's land snails, with 7% estimated by Régnier et al., (2015a) and 10–17% by Cowie et al., (2017). Based in part on these estimates of extinction proportions in land snails, Cardoso et al. (2020) suggested that 5–10% of the world's invertebrates have become extinct since the industrial age. Collen et al. (2012) provided a comparable global extinction proportion (7%) for terrestrial invertebrates; however, this was based on a very small set of 3,623 species for which Red List assessments were then available.

Our objectives in this paper are to (i) attempt to estimate the number and rate of extinctions of endemic Australian non-marine invertebrate species; (ii) describe caveats, assumptions and uncertainties around such estimates; (iii) consider how current rates of invertebrate extinctions may be addressed by, or may subvert,

current policy to prevent extinctions; and (iv) develop a set of responses that may help to reduce the current rate of extinctions in Australian invertebrates.

There are several reasons why we consider that it is desirable to estimate the tally of invertebrate extinctions: (i) it will help to provide a more comprehensive and quantitative estimate of the total loss of biodiversity in Australia; (ii) it will help to describe the consequences of historic and current conservation biases and may provide a justification for redressing those biases; (iii) it will contextualise the magnitude of the task to prevent further extinctions; (iv) it may help to identify the factors that have caused major biodiversity loss and hence those that should be managed to reduce future losses; and (v) it may help to assess and understand the likely ecological consequences of such loss. Our focus in this paper is on the extent of loss in Australian invertebrates but our findings and response are likely to be globally representative.

## Methods and assumptions

The analytical steps and assumptions are described in Tables 1 and 2, and additional details on derivation of parameter estimates and uncertainty are provided in Appendix S1.

### Number of Australian endemic non-marine invertebrate species

Our focus is on terrestrial and aquatic (i.e., non-marine) invertebrate species that are endemic to Australia. We exclude marine species because they are even less well known than terrestrial species, and because this focus makes for a more consistent comparison with extinction rates of Australian terrestrial vertebrates; we consider both terrestrial and aquatic species because many invertebrate species have life cycles that span both environments;

and we restrict the analysis to endemic species to better compare with extinction rates in Australian endemic plant and vertebrate species, and because the conservation responsibility for non-endemic species is not exclusively Australian.

The species richness of Australian invertebrates has not been well resolved (Greenslade and New, 1991; Majer et al., 2002; Yeates et al., 2003), with a small proportion of named species, and large proportions of known but undescribed species and unknown and unnamed species (New, 2022). A key requirement of our analyses was an estimate of the number of Australian endemic non-marine invertebrate species extant at the time of European colonisation. To account for inherent uncertainty in a single estimate taken from the published literature, we used five separate estimates (Table 1). These were derived from published estimates of either the Australian or global number of invertebrate species or, if unavailable, of insect species.

First, we used the estimate of Chapman (2009) of 320,465 Australian native invertebrate species. To convert this figure to non-marine invertebrate species endemic to Australia, we followed Austin et al. (2004) and Raven and Yeates (2007) recognising that at least 90% of these are endemic, and 84% of these are non-marine (Costello et al., 2012), giving an estimate of 242,272 species (Table 1).

Second, we used four independent estimates of the global number of insect species. The first three were those collated by Stork (2018) and used to derive an overall mean estimate of insect species richness: Stork (1993), Mora et al. (2011) and Stork et al. (2015), with a mean of 4.9 million insect species. Subsequent global analyses that encompass morphologically cryptic species revealed by molecular data (Larsen et al., 2017; Li and Wiens, 2023; Wiens, 2023) have led to marked increases in these tallies, with a global estimate of 21.1 million insect species (Li and Wiens, 2023). Several recent Australian studies have supported the potentially large extent of previously unrecognised cryptic diversity (Andersen

**Table 1.** Estimates of the number of non-marine invertebrate species endemic to Australia, including analytical pathway and assumptions. In the lower part of the table, the five estimates are derived by simply taking the product of each row in a column. For example, for the second column in the body of the table (Stork 1993), the estimate of the number of non-marine invertebrate species endemic to Australia is 5,900,000*1.565*0.84*(0.056—0.073)*0.9 = 390,896—509,561. The proportion of invertebrate species native to Australia is given as a range of plausible values, and as a result, most of the estimates of number of Australian endemic non-marine invertebrates are also expressed as a range; to calculate the mean of these estimates, the mid-point of the range was used

| Parameter | Assumed values (from the published literature) | | | |
|---|---|---|---|---|
| Number of insect species globally | 5,900,000 (Stork, 1993) | 2,600,000 (Mora et al., 2011) | 6,300,000 (Stork et al., 2015) | 21,100,000 (Li and Wiens, 2023) |
| Number of invertebrate species native to Australia | 320,465 (Chapman, 2009) | | | |
| Ratio of invertebrate to insect species richness | 156.5% (Chapman, 2009) | | | |
| Proportion of invertebrate species that are non-marine | 84% (Costello et al., 2012) | | | |
| Proportion of invertebrate species that are native to Australia | 5.6% (Cranston, 2009, 2010) — 7.3% (Chapman, 2009) | | | |
| Proportion of Australian invertebrate species that are endemic | 90% (Raven and Yeates, 2007) | | | |
| Number of non-marine invertebrate species endemic to Australia | 242,272 | 390,896—509,561 | 172,259—224,552 | 417,397—544,107 | 1,397,949—1,822,327 |
| Combined estimate (mean of five independent estimates, assuming midpoints of those estimates expressed as a range): | | | | 596,359 |
| Lower plausible bound: | | | | 172,259 |
| Upper plausible bound: | | | | 1,822,327 |

**Table 2.** Estimates of proportion of extinct Australian endemic non-marine invertebrate species, including assumptions

| Parameter | Lower plausible bound | Upper plausible bound | Estimate (mid-point of plausible bounds) | Assumptions |
|---|---|---|---|---|
| Proportion of Australian endemic non-marine invertebrate species extinct (1788–2024) | | | | |
| Approach 1: Proportion of extinctions in Australian endemic species of plants, fish, frogs and terrestrial reptiles, birds and mammals (1788–2024) | 0.85% | 3.10% | 1.07% | Values mostly taken from Woinarski et al. (2019, 2025). Assumes extinction proportion to be broadly similar across taxonomic groups; however lower bound excludes extinction proportion for Australian mammals, recognised to be anomalously high. See Appendix S1 for details. |
| Approach 2: Proportion of all described invertebrate species extinct in the modern period (i.e., post 1,500) | 1.40% | 2.57% | 1.99% | Assumes extinction proportions in Australian invertebrates are similar to global rates. Global extinction proportion is based on IUCN Red List assessments. Upper bound includes Critically Endangered (possibly extinct) and Extinct in the Wild species. See Appendix S1 for details. |
| Proportion of Australian extinctions (1788–2024) occurring in 2024 | 0.42% | 1.62% | 1.02% | Lower bound assumes the extinction rate has been constant. Upper bound assumes the extinction rates has been proportional to growth in Australia's human population. |

et al., 2016, 2023). To convert these four estimates to terrestrial invertebrate species endemic to Australia, we assumed that invertebrate richness is 156.5% of insect richness (Chapman, 2009) and that Australian species make up 5.6–7.3% of the global total (Chapman, 2009; Cranston, 2009, 2010). As above, we also assume that 84% of species are terrestrial (Costello et al., 2012), and 90% of Australian invertebrate species are endemic. Hence, we derived estimates ranging from 172,259 to 1,822,327 (2.5% and 97.5% quantiles matching the lower and upper plausible bounds) for the number of endemic non-marine invertebrate species (Table 1).

### Extinction tallies and rates

Extinction tallies and rates given here are for the period since the European colonisation of Australia (1788). We adopted two largely independent and complementary methods to estimate the likely proportion of Australian endemic non-marine invertebrate species that have become extinct over this period. Whenever possible, we generated lower and upper plausible bounds of our estimate of the number of extinctions.

Approach 1. The first approach was to calculate the average extinction rate as a percentage across Australian endemic species in well-known taxonomic groups (plants, freshwater fish, frogs and terrestrial reptiles, birds and mammals) (Woinarski et al., 2019, in press) and apply this proportion to the estimated number of endemic Australian non-marine invertebrate species. The principal assumption with this approach is that the extinction proportion for Australian invertebrates is similar to that of other taxonomic groups in Australia. This broad assumption has been applied in some previous assessments of the total number of global extinctions (Pimm and Raven, 2000; Dunn, 2005), although marked disparities amongst taxonomic groups in levels of imperilment and extinction are now well demonstrated (e.g., Luedtke et al., 2023). Applying the extinction proportions for Australian endemic plants and terrestrial vertebrates to Australian invertebrates is plausible, or even conservative, as there are likely to be many cases of co-extinctions of invertebrates with their plant or animal hosts (Moir and Brennan, 2020; Moir, 2021). Furthermore,

invertebrate species are more likely to have smaller ranges than vertebrate species (Yeates et al., 2002; Dunn, 2005; Harvey et al., 2011), and there is a strong relationship between range size and extinction risk (Böhm et al., 2016; Chichorro et al., 2019). Conversely, the small home ranges of many invertebrate species may allow them to persist in smaller habitat fragments than most vertebrate species could. Invertebrate species are likely to be susceptible to many of the same factors that have been responsible for extinctions in other taxonomic groups (Sands, 2018; Cardoso et al., 2020), in addition to threatening processes that have limited impacts on plants and vertebrates (such as widespread use of insecticides) (Dunn, 2005; Sands, 2018; Samways et al., 2020; Wagner et al., 2021). Furthermore, the proportion of unrecognised and undescribed species is far higher for invertebrates than for plants and vertebrates, and there is a tendency for higher rates of loss and imperilment amongst undescribed than described species (McKinney, 1999; Liu et al., 2022; Boyle et al., 2024). Also, there is a strong bias in conservation response and investment towards vertebrates, particularly mammals and birds (Walsh et al., 2013), so that imperilled species in these groups would have been more likely to have been saved from extinction through conservation investments and actions than for equally imperilled invertebrate species (Langhammer et al., 2024). Given these characteristics and assumptions, the application of the extinction rates for Australian vertebrates and plants is likely to be conservative for estimating the proportion of Australian invertebrate extinctions. However, the extinction rate for Australian mammals is exceptionally high relative to other taxonomic groups in Australia, and relative to mammals globally (Woinarski et al., 2015), so in the lower bound strand of this analysis, we recognise this atypicality and exclude mammals from the averaged extinction rate applied to Australian invertebrates.

Calculations of extinction rates in Australian plants and groups of vertebrate animals are detailed in Appendix S1. The average extinction rate across Australian plants, and Australian endemic freshwater fish, frogs, terrestrial reptiles, birds and mammals is 1.07%; the lower bound (excluding the exceptional rate for mammals) is 0.85%, and upper bound is 3.10%.

Approach 2. We used the conservation status assigned by the IUCN to invertebrate species (IUCN, 2023) and applied the global percentage of extinct invertebrate species to the estimated number of Australian invertebrate species. This method assumes Australian invertebrates have become extinct at the same rates as invertebrates globally. This assumption may be tenuous because, for most taxonomic groups, threats and rates of decline and loss vary globally. Extinction rates are particularly high on islands (Régnier et al., 2015b; Terzopoulou et al., 2015; Yeung and Hayes, 2018; Cowie et al., 2022), and Australia has many islands (> 700 with area > 1 km$^2$), including Tasmania and endemic-rich oceanic islands such as Christmas, Lord Howe and Norfolk (Woinarski et al., 2018; Hyman et al., 2023), and the long period of Australia's isolation has also meant that the Australian biota shares the island characteristic of susceptibility to new threats (Woinarski et al., 2015; Legge et al., 2023). Furthermore, Australian rates of habitat destruction and fragmentation have been above global averages, for example, with >40% forest loss (Bradshaw, 2012), and many invasive species now occur pervasively across the continent (Legge et al., 2017).

The IUCN assessments we used (IUCN, 2023) cover the period 1,500–2023, whereas our interest here is in extinctions since 1788; however, this different timespan is unlikely to have a substantial impact because very few of the recognised global extinctions occurred in the period 1,500–1788 (Ceballos et al., 2015). The IUCN Red List status assessments of invertebrates (27,363 species) encompass only a small proportion of the world's invertebrate species (>7 million: Stork, 2018), and it is possible that attention has focused particularly on those invertebrate groups known to be particularly imperilled, which may over-inflate the extinction proportion. Conversely, IUCN assessments are mostly undertaken only for described species, with undescribed species assessed only in exceptional circumstances, so the likely many cases of dark extinctions of invertebrates are heavily under-represented, giving conservative estimates. The IUCN Red List categories include extinct in the wild and Critically Endangered (possibly extinct). The 'possibly extinct' label is used by the IUCN as a tag to denote species that are likely already extinct (or extinct in the wild) but require more investigation for this to be confirmed. As an upper bound, we add these to the global tally of extinct invertebrate species. Of the 27,363 invertebrate species globally for which conservation status has been assessed by the IUCN, 384 (1.40%) are listed as extinct, and a further 320 species are considered extinct in the Wild or Critically Endangered (possibly extinct) (collectively 2.57%). The mid-point of these rates is 1.99%.

### Annual rate of extinctions: Prediction for 2024

To predict the current (i.e., 2024) annual rate of extinctions from the estimated tally of extinctions over this 236-year period, we took two alternatives: (i) assume that the annual extinction rate is constant over this period or (ii) assume that the extinction rate is variable over time and related to the cumulative extent of environmental modification, here represented by changes over time in the size of Australia's human population (Figure S5). The constant annual rate is conservative and unlikely, as the threat burden on invertebrates was undoubtedly far less in earlier years than in recent years (Régnier et al., 2015b), although a reasonably constant rate of extinctions (at least since about 1840) was reported for the set of 97 Australian extinctions described by Woinarski et al. (2019). For the latter approach, assuming the invertebrate extinction rate mirrors the growth in the human population size, the expected number

of extinctions in 2024 is 1.62% of all extinctions over the period 1788 to 2024 (Figure S6). This second approach is more plausible, but human population size is a very inexact surrogate for threat load. In reality, extinction rates have probably varied over time with pulses of extinctions of Australian invertebrates associated with the introduction of rodents to oceanic islands, episodes of intensive habitat destruction, fragmentation and consequent extinction debt and, increasingly, impacts from climate change (Harvey et al., 2023; Wiens and Zelinka, 2024), albeit perhaps moderated by increasing establishment of the conservation reserve system and other conservation management and legislation. In analyses below, we adopted the uniform rate of extinctions over time as the lower bound for estimating the proportion of post-1788 extinctions that will occur in 2024 (i.e., 0.42% [= 1/236] as the current annual rate). For the upper bound, based on the assumption that the extinction rate varies over time in parallel with human population, we determined that 1.62% (i.e., Australia's population in 2024 (26.7 million) divided by the sum of annual tallies of Australia's population across the years 1788 to 2024) of the total number of extinctions since 1788 will occur in 2024.

### Analysis

For each of the two approaches to estimating extinction rates, we use Monte Carlo simulation to make 100,000 choices of combinations across the simulated distributions of the two initial elements in the chain of analysis (numbers of Australian non-marine invertebrate species, proportional extinction rate). To derive the two simulated distributions, we assume that our lower and upper estimates correspond to 95% confidence intervals (see Appendix S1 for workings). We then estimate the numbers of extinctions expected in 2024 from the outcomes of this Monte Carlo simulation, based on the extinction rate being constant over years or on the annual rate being concordant with growth in the human population.

### Results

Detailed results are given in Appendix S1 and summarised in Table 3. We estimate that since the European colonisation 236 years ago, the number of Australian endemic non-marine invertebrate species rendered extinct is 9,111, with 2.5% and 97.5% quantiles matching the lower and upper plausible bounds of 1,465—56,828 (Table 3). This overall estimate

**Table 3.** Estimates of the total number of extinctions of Australian endemic non-marine invertebrate species since European colonisation (1788—2024), and in 2024 alone

| Number of extinctions | Estimate | Lower plausible bound | Upper plausible bound |
|---|---|---|---|
| Since European colonisation (1788—2024) | | | |
| Approach 1: Australian extinction rate (non-invertebrates) | 6,367 | 1,465 | 56,828 |
| Approach 2: Global extinction rate (invertebrates) | 11,856 | 2,402 | 47,133 |
| Combined estimate | 9,111 | 1,465 | 56,828 |
| In 2024 | 39—148 | | |

combined two approaches: Approach 1 (using the Australian extinction rate for non-invertebrates) suggested 6,367 species (plausible bounds: 1,465—56,828) and Approach 2 (using the global extinction rate for invertebrates) suggested 11,856 species (2.5% and 97.5% quantiles: 2,402—47,133). These tallies equate to an expected number of extinctions occurring in the year 2024 of 39—148 species, equating to around 1—3 extinctions per week.

## Discussion

Our estimate of the number of Australian non-marine invertebrate species that have become extinct since 1788 (about 9,100 species) recalibrates long-held perceptions of the extent of biodiversity loss in Australia and its taxonomic characteristics. Our estimate vastly exceeds the extinction tally of 97 species reported across all taxonomic groups in Australia (Woinarski et al., 2019), and the single species of invertebrate listed as extinct under Australian legislation. Whereas almost all extinctions of Australian vertebrate species have been formally recognised and hence dominate the extinction narrative, we conclude that only a tiny proportion (ca. 0.1%) of the invertebrate extinctions have been recognised, and only about 0.01% of the invertebrate extinctions are listed under Australian law. This indicates a massive distortion and under-appreciation of the historic and ongoing loss of Australian biodiversity (Figure 1).

But, even more importantly, our analysis provides a warning of the likely continuing and escalating high rates of looming extinctions. We predict that 39—148 Australian endemic non-marine invertebrate species will become extinct in 2024 (i.e., 1—3 extinctions per week) and that unless there is a major increase in investment and change in conservation priorities, and more effective control of threats, this rate of extinction will increase. We should not simply maintain the current conservation *status quo* and let these extinctions happen. Our assessment should provide a catalyst for redressing some of the taxonomic inequality in conservation.

Are these tallies plausible? We explicitly recognise many assumptions and caveats in these estimations; however, most of these assumptions are conservative. The two main lines of evidence that we use to estimate the proportional extinction rate since 1788 of Australian endemic invertebrate species are based on independent approaches but result in tallies that are of comparable magnitude. Our estimates are based on proportional extinctions for Australian invertebrate species of 0.9–3.1% (Approach 1) and 1.4–2.6% (Approach 2) (Table 2), substantially lower proportions than the global estimate of 5–10% assumed by Cardoso et al. (2020). We recognise the wide bounds around our estimates but consider that these are currently inescapable particularly given uncertainties about the total number of invertebrate species. However, even the low bound of our estimate represents a vast increase in previously reported numbers of extinctions in Australia.

Extinction in Australian invertebrates has undoubtedly fallen unevenly across taxonomic groups. Characteristics of some invertebrate groups render them particularly susceptible to extinction (New, 2022; Harvey et al., 2023). Many Australian non-marine invertebrates are short-range endemics (< 10,000 km², with some known only from a single site or a habitat patch of a few hectares) (Moir and Young, 2023), with such range limitation associated with limited dispersal capability, susceptibility to disturbance and desiccation and extreme habitat specialisation. Examples include many species in groups such as Heliozelidae (micromoths), Triozidae (plantlice), Gastropoda (snails and slugs), Oligochaeta (earthworms), Onychophora (velvet worms), Araneae (mygalomorph spiders), Diplopoda (millipedes), Phreatoicidea (phreatoicidean crustaceans) and Decapoda (freshwater crayfish) (Harvey, 2002). Such species are particularly at risk with even small extents of habitat destruction or degradation. Furthermore, even

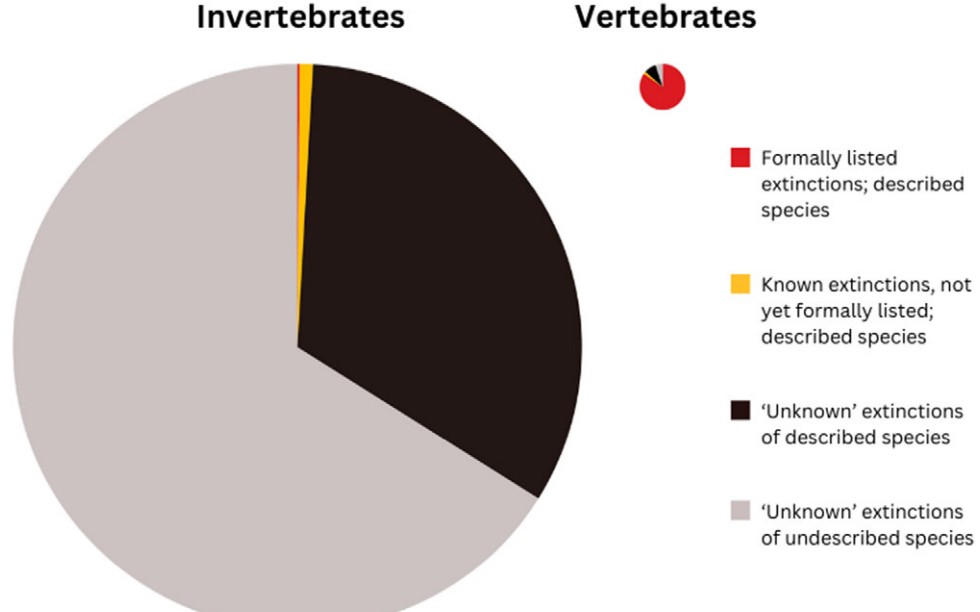

**Figure 1.** Schematic representation of the numbers of Extinct Australian endemic vertebrate species (right) and invertebrate species (left), drawn to approximate scale (i.e., the ratio of the pie area for invertebrates to that of vertebrates is similar to the estimated ratio of number of extinctions). 'Listed' means formally recognised as extinct by the IUCN or under Australian legislation. The four categories in each pie represent the number of formally recognised and listed extinctions (red); the number of known extinctions, that are not yet formally listed, of known species (yellow); the estimated number of 'unknown' extinctions of described species (black); and the estimated number of 'unknown' extinctions of undescribed species (grey).

where their habitat is protected within conservation reserves, short-range endemics with specialised habitat requirements may be at risk from other threats such as weeds, disease, fire and introduced animals. For example, for such short-range endemics, fires of exceptional severity, or where the interval between fires is too short to allow for recovery may eliminate the entire population (Gibb et al., 2023); or, if occurring at more than one site, the consequences of any such local losses are likely to be compounded by their typically poor dispersal ability reducing the likelihood of recolonisation from any patches that may have evaded destruction. Island endemics represent a particular type of short-range endemism, as long periods of isolation have left them with pronounced susceptibility to invasive species. Extinction risk is also high for invertebrate species with tight dependency on other species (Moir, 2021). Species occurring in environments that are now extensively modified or transformed (e.g., temperate grasslands now largely used for intensive agriculture or housing developments) may also be at high risk of extinction. Increasingly, climate change will accentuate susceptibility, compound the impacts of existing threats, and ratchet up extinction rates (Harvey et al., 2023; Wiens and Zelinka, 2024). Species associated with particularly narrow climatic or biotic envelopes (e.g., karstic or montane refugia) may be particularly likely to disappear as their habitat and ecological framework is subverted by climate change.

Our results serve to illustrate the consequences of the taxonomic biases permeating conservation. Although surveys have shown that the Australian public supports commitments to prevent extinctions (Zander et al., 2021, 2022), relative to vertebrates and plants, invertebrates are valued less by the Australian community (Tisdell et al., 2007), so there is less concern for their conservation and relative indifference to their extinction (Woinarski et al., 2024). As a consequence of such pervasive taxonomic biases, conservation of the Australian (and global) invertebrate fauna is hampered by profound knowledge gaps in taxonomy, distribution, threats, life cycles, ecological interactions, management needs, population size and trajectory and conservation status (Cardoso et al., 2011b; Taylor et al., 2018). Yet, this knowledge is critical for listing species as threatened (and hence providing them with some conservation protection and scrutiny) and for management to prevent extinction. In part because of public disinterest, governments invest less in the conservation of imperilled invertebrates: for example, Australian governments allocated at least $12 million towards the recovery of the Endangered Koala, *Phascolarctos cinereus*, following high severity wildfires in 2019–2020 (which burnt 17% of koala distribution), far more than the collective investment in recovery for 382 invertebrate species that had all of their known range burnt (Marsh et al., 2022). Furthermore, some ecologists have argued that the extinction of any invertebrate species is typically less consequential than for a vertebrate species, on the presumption that there is much more ecological redundancy amongst invertebrates (Walker, 1992). Such biases and knowledge gaps all serve to render invertebrate species increasingly imperilled; make it less likely that efforts are made to prevent their extinction; less likely that any such efforts, if made, will be successful; and, when invertebrate extinctions occur, less likely that they are noticed, formally recognised or mourned.

Australian government conservation policy now has stated aims to prevent extinction (Commonwealth of Australia, 2022) with an explicit objective over a 10-year timeframe (2022–2032) that "new extinctions of plants and animals are prevented" and a target that "species at high risk of imminent extinction are identified and supported to persist". This is explicitly (and admirably) egalitarian: all species are meant to be covered by this protection. With a plausible, and likely undiminishing, rate of 1–3 extinctions of Australian invertebrates per week, this commitment is clearly not being met, and can be met only if highly imperilled invertebrates are recognised and supported. However, so long as invertebrate extinctions remain nameless and invisible, this failure cannot be demonstrated, or readily overcome; and efforts will instead be directed towards the less imperilled, but better-known and iconic species.

To some extent, our assessment is clutching at air: although we can estimate the likely number of extinctions, we cannot put names to (most of) them. This invites scepticism; and the anonymity of the extinct species may simply reinforce public and political disinterest and incredulity. Whereas tangible evidence is available for some dark extinctions – for example, diagnostic shells of some land snails may persist long after the species has disappeared (Régnier et al., 2015b) – many extinct invertebrate species are likely to have disappeared and left no trace: they were never discovered and will never be so now. We coin the term 'ghost extinctions' for such cases of dark extinctions where, in the absence of any physical evidence, the likely existence and subsequent loss of a species may be imputed solely from ecological, evolutionary or taxonomic reasoning. For example, for the isopod genus *Crenoicus*, Wilson (2008) noted the known extinction of one Australian species, that sampling had been limited across the geographic range of the genus, that there was a high level of speciation and short-range endemism, that there was a tight dependence of extant species upon naturally fragmented environmental features (highland springs and Sphagnum bogs) that were now largely cleared or heavily degraded; and, on this basis, argued that many (undiscovered and now undiscoverable) species were likely to have existed but were now extinct.

Our analysis provides estimates of the number of extinctions and rates of ongoing extinctions in Australian invertebrates. However, the main impediment to preventing further extinctions of Australian invertebrate species, or even in reducing the rate of extinctions, is that mostly we do not know which species most need help to prevent extinction (most will not even be described: Figure 1), where they are, or what help is needed.

Notwithstanding the difficulties, there are recognised approaches that can foster better conservation outcomes (including constraining the rate of ongoing extinctions) for invertebrates in Australia, and globally (Sands, 2018; Taylor et al., 2018; Harvey et al., 2020; Kawahara et al., 2021; Braby et al., 2021; New, 2022). Foundational to such conservation change is recognition of the current rate of invertebrate declines and extinctions and the ecological ramifications of such extensive loss, including subversion of the numerous and pervasive ecological services, many vital for our existence, provided by invertebrates (Cardoso et al., 2020; Samways et al., 2020). Also fundamental is the need to increase public and political awareness of invertebrates, including their values and rights (Woolaston and Akhtar-Khavari, 2020); and that community concern for, and government commitments to, preventing extinction should better encompass invertebrate species.

More investment is needed to increase knowledge about (and hence increase the capability to conserve) invertebrates, especially in taxonomy, but also inventory, monitoring and identifying the key threatening processes that affect them. Some streamlining of knowledge gain is possible (Costello et al., 2013): for example, advances in barcoding and e-DNA sampling are allowing quicker and more comprehensive inventory, monitoring and species recognition (Ruppert et al., 2019; Liu et al., 2020), and such advances could be used to underpin a national monitoring programme that can encompass currently undescribed species. However, even with such technological progress and significantly more investment, the

rate of knowledge gain may well not match the rate of biodiversity loss, so additional conservation approaches are also needed (Moir and Brennan, 2020). One priority would be to attempt to consolidate existing museum collections that already preserve vast numbers of undescribed species and collect as comprehensively as possible across currently undescribed species that have yet to be sampled to maintain some record of species likely to become extinct in the near future. At least then, future generations may have some appreciation of what has been lost (Cowie et al., 2022). A national programme dedicated to species discovery, BushBlitz, has been operating since 2010 and has detected more than 1,900 new invertebrate species (https://bushblitz.org.au/).

As an additional conservation approach, we suggest that Australian specialists develop a collated list of potentially extinct invertebrate species, including 'lost' species unreported for many decades (e.g., Hyman et al., 2023). This may help add substance to our estimated extinction tallies. But even more importantly, if such species are not actually extinct, they may be highly imperilled and may need prioritised conservation attention. Such lists of lost species have been developed for vertebrates, prompting public interest, targeted searches, and consequently, in some cases, rediscovery and the urgent implementation of conservation actions needed to prevent extinction (e.g., Lindken et al., 2024). Indeed, there are several cases of Australian endemic terrestrial invertebrates that were thought to be extinct (based on long periods without records) that have been re-discovered recently as a consequence of further targeted searches: examples include the flea *Wurunjerria warnekei* (Steventon et al., 2022) and the beetle *Cormodes darwini* (Reid and Hutton, 2019). Developing a better inventory of probable extinctions also helps respond to the plea of Dunn (2005): "If we are serious about insect conservation, we need to spend more time and money documenting such extinctions".

Currently, most imperilled Australian invertebrate species are not given the explicit protection and conservation priority that flows, at least in principle, from formal listing of species as threatened, because the available knowledge of their status falls below the evidentiary bar required for listing. We recommend more use of the precautionary principle in such cases, to allow for the listing of poorly known species at high risk of extinction, the inclusion of co-dependent species when listing better-known threatened species (Moir and Brennan, 2020), tailoring listing criteria such that poorly known imperilled species are not so readily excluded (Cardoso et al., 2011b), and more use of listing of threatened ecological communities that encompass imperilled invertebrate species (Taylor et al., 2018). However, we note that major increases in nominations for threatened species listing of a substantial proportion of imperilled invertebrates may exceed the resources currently assigned by governments to the listing process. Furthermore, the addition of many more species to the threatened species list will overwhelm the already insufficient budget available for implementing conservation actions (Wintle et al., 2019); and we anticipate that a vast increase in the number of formally recognised threatened species may be unpalatable to governments. However, these are arguments for wilful neglect and for failing to invest sufficiently in conservation and are inconsistent with stated objectives for preventing extinctions.

The precautionary principle should also be used more widely in the assessment of potential impacts of development proposals upon poorly known species. For example, under Western Australian (but not national) policy, development proponents need to undertake comprehensive sampling of biodiversity at a site, and then demonstrate that any potentially affected species also occurs elsewhere (Environmental Protection Authority, 2009).

In addition to conservation focus for individual imperilled invertebrate species, a network of 'coarse filter' conservation responses is required to better represent the conservation and recovery needs of large and diverse assemblages of imperilled invertebrate species and invertebrates generally (Samways et al., 2020). Such actions include improvements to policy and legislation to provide more effective constraints on habitat destruction, emission of greenhouse gases and use of pesticides; habitat restoration and reconnection; improved fire management; more effective biosecurity (such as enhanced quarantine standards for detecting entry of potential invasive species at oceanic islands of conservation significance, and for Australia generally, increased surveillance aimed at early detections of incursions, and commitments for adequate resourcing to eradicate such incursions); and increase in the extent and comprehensiveness (and improved management) of the conservation reserve system (Chowdhury et al., 2023). Some of these conservation actions are occurring: one notable example is the recent eradication of introduced rodents from Lord Howe Island, which will reduce the extinction risk for many highly imperilled invertebrate species (e.g., Reid and Hutton, 2024). Short-range species are likely to have comprised much of the losses of invertebrates to date and to be most susceptible to future loss. The distributions of many of these species co-occur at finer- and coarser-scale centres of endemism (Harvey, 2002; Moir et al., 2009, 2016; Eberhard et al., 2009; Murphy et al., 2013; Gibb et al., 2023; Moir and Young, 2023), and the effective conservation of such areas may avert many otherwise likely extinctions. Such areas need to be identified, included in the reserve system and managed to control threats.

Because they are largely unnoticed and unmourned, Eisenhauer et al. (2019) deemed losses of invertebrate species as 'quiet extinctions'. We paraphrase this lack of resonance in our title, 'This is the way the world ends; not with a bang but a whimper' taken from the final lines of TS Eliot's poem, The Hollow Men. The expression is apt also as, given the foundational role of invertebrates in ecological systems, the cumulative losses of seemingly inconsequential invertebrate species are likely to have led, and continue to lead, to far-ranging ecological effects and ecosystem subversion, and consequently to impacts on productivity and human health (Cardoso et al., 2020).

**Open peer review.** To view the open peer review materials for this article, please visit http://doi.org/10.1017/ext.2024.26.

**Supplementary material.** The supplementary material for this article can be found at http://doi.org/10.1017/ext.2024.26.

**Data availability statement.** Relevant data are given in Appendix S1.

**Acknowledgements.** We thank Dr. Libby Rumpff for comments on policy implications. We are also most grateful for the helpful comments of four anonymous reviewers.

**Author contribution.** JCZW conceived this study, contributed to analysis, and wrote an initial draft. BPM undertook the analyses. MLM contributed to analysis. All authors contributed to writing.

**Financial support.** No specific funding support.

**Competing interest.** The authors declare no conflict of interest.

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
