## [Editor Report · Recommendation: 
*This is the way the world ends; not with a bang but a whimper*: Estimating the number and ongoing rate of extinctions of Australian non-marine invertebrates — R0/PR2]

This is an important topic and a well-developed study, with potentially valuable implications for conservation, but there is a number of issues identified by the four reviewers that should be addressed before the manuscript can be considered suitable for publishing. I found especially relevant suggestions by the second reviewer to provide more information about the analysis and to emphasize better limitations of the study throughout the manuscript.

Please note that the first reviewer has also provided additional comments in the enclosed manuscript document.

1. Reviewer 1:

1.1. This submission is a valuable contribution to the increasing understanding of the true calamity of global biodiversity extinction and the fact that many species of invertebrates are going extinct without us even knowing the species existed - hence ‘dark’ extinctions and now ‘ghost’ extinctions, which I like. The estimates of the level of extinction in Australia are very much in line with other recent estimates (eg. those generated and others briefly reviewed by me and my colleagues: Cowie et al 2022, Biological Reviews). Some of the numbers are incredulously high at first sight but as more such estimates are published I think a consensus is being generated that is quite horrifying. The quantitative part of this manuscript is one such extremely valuable and rigorously generated addition (albeit with lots of inevitable caveats, as for other previous estimates). The discussion covers ground that has been covered more than a few times before one way or another. Nonetheless, it is prompted by the new Australian estimates presented and deserves to be published as a clarion call to the Australian authorities (that sadly will probably go largely ignored) as well as as grist to the mill internationally regarding the ongoing loss of biodiversity that is being largely ignored.

1.2. I would add one thing that the authors might consider adding to their discussion. Assuming pessimistically that the trend of extinction will continue, then the key is to collect as many novel invertebrate species as possible and preserve them in museums. This is even more important than a focus on describing and naming them, which might take 1000 years or more, but at least it would mean that people in the future could know what had been lost. Without this effort, but a focus on naming/describing, although we would describe/name a bunch of species, many more would still become ‘ghost’ extinctions.

1.3. I have no major criticisms of the manuscript but I have marked up a Word version of the manuscript with minor comments that the authors should address. Apparently I am unable to attach this document to this review, so I will send it to the editorial office for transmission to the authors. 

2. Reviewer 2:

2.1. This was a difficult paper to review. I appreciate the motive and goal; of course we need more science to advocate for better insect conservation policies in Australia, which is clearly what the authors are aiming to provide. But, as the authors note explicitly and implicitly throughout, it is nearly impossible to determine an accurate estimate of the number of Australian invertebrates already lost and at risk of extinction, given the ridiculously noisy and absent data for this group of animals. Analyses like these, as statistically/technically appropriate as they may be, cannot truly be verified. To quote the authors, it is ‘clutching at air’. 

2.2. Philosophically, one could question should an analysis such as this, with extremely high levels of uncertainty that can’t be adequately addressed, be published at all, given the predictable media coverage and the potential to fuel misinformation about scientific processes and insect ecology generally. But some would argue that any estimate is better than none to start a conversation. Perhaps this is true, if the analysis is presented with reasonable acknowledgement of limitations. 

2.3. My main comments relate to the broader conceptual and theoretical framework for the study, rather than technicalities of the analysis. In particular, I think the paper is missing ecological depth and I think embedding more ecological knowledge in the analysis, interpretation and discussion would improve the rigour of the estimates. I also suggest that the authors consider resubmitting this as a Perspective, which may be a more suitable category for the content than a Research Article. 

2.4. The authors appear to have taken a reasonable coarse-scale approach to estimate results with the very limited and biased data available. There are numerous caveats, many of which can’t be adequately addressed with available data, including the huge uncertainty around numbers of species, the taxonomic biases and limitations of the IUCN data, and the inability to accurately predict population dynamics, and therefore extinction risk, for the majority of invertebrate species. To be fair, the authors acknowledge many limitations in the Discussion (which very few general readers will see), although this doesn’t come across in the Abstract and Impact statement (which will arguably get the most attention). I would like to see some stronger statements in these sections that unambiguously clarify the limitations and caution against misuse of these estimates.

2.5. In particular, given the huge range of the estimates, it seems scientifically questionable to provide a single extinction rate/number of species with such confident statements, e.g. 9111 species, 1-3 species going extinct per week. For people not familiar with the data limitations and statistical methods (i.e. most general readers and news consumers), there is very little context available to interpret how ‘accurate’ these statements really are, and given that these are the summaries that will be plucked from the abstract and spread widely through media platforms, I think the authors could make more effort to present these estimates more responsibly.

2.6. A key issue with estimating invertebrate extinction rates is the huge variation in life history, biology and ecology of invertebrate species. This makes it particularly difficult to translate extinction rates from vertebrates and plants under the assumption that these rates will apply similarly across the range of invertebrate groups. I appreciate that the authors are attempting to make a coarse-scale estimate for policy purposes which doesn’t account for the ecological nuances of different groups. But I think the authors could provide some more nuanced discussion of how extinction risk could vary at e.g. family/order, or even class, level.

2.7. Another key assumption that I find problematic is the assumption that any invert species that hasn’t been recorded for a while, or inhabits a restricted area, is therefore ‘highly imperilled’ or at high risk of extinction. This assumption is woven throughout the manuscript, presented both explicitly and implicitly through text and analyses. I appreciate the sentiment, and from a conservation policy perspective it might be useful to assume this. But there are so many factors at play, including the funds, resources and expertise that limit people out looking for invertebrates, the seasonal and environmental conditions that drive detectability for many species, dispersal ecology, generalist vs specialist species etc….there are robust lines of evidence to also assume that invert species ranges may be much larger than we know and that many invert species may be able to adopt strategies that avoid or minimise extinction when faced with a new disturbance or threat. Both assumptions are scientifically valid, yet the argument in this article seems more skewed toward the first assumption – I think greater balance on the nuances of invertebrate ecology is needed throughout.

2.8. Line 100: it’s a disservice to science that the Sanchez-Bayo & Wyckhuys papers continue to be cited as evidence of a phenomenon that they do not have the methodological capacity to provide evidence for

2.9. Line 155-190 and Table 1: The process used to calculate estimates of invertebrate diversity is confusing to follow. I appreciate the attempt to use multiple different sources to gather a range of estimates. I didn’t have time to crosscheck every single reference cited here, although some of the publication years are quite old and there may be some more up to date refs for some of these calculations. In particular, it would have been good to see some more ecologically sound estimates based on currently available taxonomic lists of Australian species, rather than assumptions and extrapolations from broader global estimates. The Australian faunal directory lists numbers of species, and although not fully complete it is the closest to a currently accepted list of known species. Working through the Animalia checklist, I got a total of 115,703 known invertebrate species – given estimates that only 30% of our species are described, this gives an estimate of approx. 385,680 species. And about 90% would be terrestrial/freshwater (this was a quick calculation based on my knowledge of the general ecology and life history of each phylum, or subgroups where only part of the phylum are marine – I didn’t spend too much time on the smaller/more obscure groups so may have missed a few species) https://biodiversity.org.au/afd/mainchecklist. I’m not suggesting the authors need to use these estimates, just highlighting an example of how including ecological knowledge may improve the rigour of the estimates. 

2.10. Line 185: I didn’t try and back-calculate this, but I’m not sure where the 156.5% comes from – can the authors please explain this more clearly?

2.11. Line 292: Please cite the statistical software/platform used for analysis

2.12. Line 345-369: lots of discussion about taxa that are more susceptible to extinction, suggesting these are common traits across all inverts. Would be appropriate to also acknowledge what groups/traits are less susceptible to extinction, variation across groups, and how this relates to limitations and influences results.

2.13. Line 399: I would argue it’s not just the ‘highly imperilled’ inverts that need to be recognised and supported to meet these aims...

2.14. Line 450: what evidence is there to suggest that these species are “likely” to be highly imperilled? 

2.15. Line 455-456: Numerous examples of inverts found many years after thought to be extinct, mostly because of detectability issues - Lord Howe Island stick insect is an obvious example, also Lord Howe Is cockroach, Norfolk Island snails, Key’s matchstick grasshopper, Tasmanian short-tailed rain crayfish, Douglas’ broad-headed bee in Perth…etc

3. Reviewer 3:

3.1. This work intends to estimate the number of extinctions for Australian invertebrates since European colonization, reaching staggering numbers for both historical and contemporary (2024) extinctions. The numbers are alarming and convey an extremely important message. I only have a few comments to the text.

3.2. Abstract, line 24: I would say the consequence of pervasive taxonomic biases in knowledge is not the high rate of loss but the lack of recognition on which species were effectively extinct?

3.3. I think mentioning Sánchez-Bayo and Wyckhuys 2019 should be avoided as the paper is plagued with errors and does not really meet reasonable scientific standards

3.4. Ln 234 - I wonder if rate is the appropriate word throughout the manuscript? This is a sum, not a change per unit of time, which I think is the proper use of rate. But I am no native speaker.

3.5. In methods, and at the first time lower and upper bounds are reported, mention these are the 2.5% and 97.5% simulated values, at first I was thinking about min and max.

3.6. Ln 322 – 87 out of 9000 is 1%, not 0.1%?

3.7. Ln 348 – To avoid confusion with the specific meaning of location in the IUCN Red List, maybe this term can be replaced by locality or similar?

4. Review 4:

4.1. It is such a relief to see that this paper exists and will be published. The logic is well set out and all assumptions and their consequences clearly explained. This paper will probably be controversial, but healthy debate around this topic will encourage action that may start to address many of the knowledge gaps the paper mentions. We must start where we are, and this paper does an excellent job at describing as best we can where we currently are. Estimating extinctions with the data we have in terms that help highlight the risk to Australia in continuing business as usual for biodiversity conservation resourcing is an important step in the right direction. 

4.2. It has a strong taxonomic flavour most likely due to the expertise of many of the authors. It could be interesting to touch more on ecology, evolutionary, applied and human health angles when discussing the impact and problems. 

4.3. I would drop the use of non-charismatic. The more this gets repeated, that invertebrates are not charismatic, the more people accept it to be true. In fact, one could posit that it is interesting that these extinctions and the neglect of this group continues to go unaddressed despite the charisma of many invertebrates, like butterflies, dragonflies, giant squid, octopuses, starfish and their many amazing relatives.

---

## [Editor Report · Recommendation: 
*This is the way the world ends; not with a bang but a whimper*: Estimating the number and ongoing rate of extinctions of Australian non-marine invertebrates — R1/PR5]

As suggested by both reviewers, the authors have adequately addressed all their comments. One of the reviewers has suggested some further minor edits, made directly in the enclosed manuscript document. Once these comments have been addressed, the manuscript should be suitable for publishing.

Reviewer 1:

I created a Word file from the pdf provided to me and used track changes to make comments and suggest changes/corrections, all minor and rather few of them. This is a very worthwhile paper and I look forward to seeing it published. As before, the system does not seem to let mu upload an attachment, so I will send my marked-up manuscript to the editorial people to forward to you. 

Reviewer 2:

The authors have argued their responses appropriately.

---

## [Editor Report · Recommendation: 
*This is the way the world ends; not with a bang but a whimper*: Estimating the number and ongoing rate of extinctions of Australian non-marine invertebrates — R2/PR8]

Authors have closely followed reviewers' suggestions and adequately revised the manuscript. I think that the manuscript can be now accepted for publication.